# "Miles Is a Mode; Coltrane Is, Power": Notes on John Coltrane as Poetic Muse and Michael Harper's "Alone" in *Songlines in Michaeltree* (2000)

**James Mellis**

Guttman Community College, New York, NY 10018, USA; james.mellis@guttman.cuny.edu

**Abstract:** This article looks at the ways jazz legend John Coltrane was a muse for many Black Arts era poets and proceeds to discuss how Michael Harper rendered Coltrane in his work, focusing on editorial changes between the 1970 and 2000 versions of Michael Harper's poem, "Alone". In it, the author argues that the change marks a revision of the centrality of Coltrane as Harper's muse from his early to later career.

**Keywords:** Michael Harper; African-American literature; poetry; jazz; John Coltrane

When John Coltrane died of liver cancer in 1967, he left behind a musical and cultural legacy that has been subject to innumerable musical, cultural, and literary studies. During the four short decades of his life, Coltrane embarked on a musical evolution that witnessed him passing through a series of professional permutations, including but not limited to: Charlie Parker acolyte; sideman to Dizzy Gillespie, Miles Davis, and Thelonious Monk; obsessive, innovative bandleader "chasing the Trane", while creating "sheets of sound"; and the progenitor of a free-jazz that seemingly sought to transcend all musical constraints and engage in conversation with both his unconscious and the universe writ large.

Concurrently, John Coltrane seemingly inhabited a number of personal identities, such as student of musical theory; alcohol- and heroin-addled addict; obsessive music practitioner; loving father and husband; and spiritual seeker writing poetic Psalms, while exploring spirituality in his music declaring, "I want to be a saint", and chanting from the Bhagavad Gita. Or, as Henry Lacey declares in one expansive summation of Coltrane's legacy in "Baraka's Am/Trak: Everybody's Coltrane Poem", the musician was, to many, a "consummate craftsman, expositor of the Black past, musician–artist-priest", who attempted to "serve as a healing force in a decadent land" (Lacey 1986, p. 18).

Upon his death, which took many by surprise, a process of poetic tribute and veneration that had already begun during his lifetime (as seen in A.B. Spellman's 1965 poem "John Coltrane" and Michael Harper's "Dear John, Dear Coltrane", which was written in 1966, though not published until 1970) quickly and rapidly escalated. Coltrane's death, occurring within the short span of years that also witnessed the deaths of Malcolm X, Martin Luther King Jr., and so many other prominent Black political and cultural figures, places him in the midst of a pantheon of martyr–rebels of the late 1960s, which allowed writers, particularly poets of the Black Arts era, to adopt and romanticize the figure of Coltrane and, through their tributes, re-imagine him in any number of ways, an effort that reached an early apotheosis in John Taggart's 1969 *Maps #3: Poems for John Coltrane*, a volume dedicated solely to poems about Coltrane (three of which were written in Spanish).

That Coltrane's cultural legacy exists beyond his music to poems that honor, reimagine, and pay tribute to him, is readily apparent. Indeed, according to Sascha Feinstein in the preface to *The Jazz Poetry Anthology*, "John Coltrane has probably been the focus of more poems than any other jazz musician, but the portraits of the man and his music vary as much as his own creative endeavors" (Feinstein and Komunyakaa 1991, p. xix).

This assessment is echoed by Samo Šalamon in "The Political Use of the Figure of John Coltrane in American Poetry", who notes that "The Coltrane Poem became a genre not only in African American poetry, but in American poetry in general, and also in poetry on other continents" (Šalamon 2007, p. 82), and Kimberly Benston, who writes in *Performing Blackness: Enactments of African-American Modernism*, "The 'Coltrane Poem' has, in fact, become an unmistakable genre of contemporary black poetry . . . and it is in this genre that the notion of music as the quintessential idiom, and the word as its annunciator, is carried to its technical and philosophic apex" (Benston 2000, p. 120).

Some critics, such as Gerald Early in "Ode to John Coltrane, A Jazz Musician's Influence on African American Culture", wonder as to the ubiquity of the Coltrane poem, pondering, "why this particular jazzman should have become such an iconographic figure" (Early 1999, p. 371). Early proceeds to answer this question (after a rather critical assessment of Coltrane's spirituality), by pointing to the increasing popularity of jazz in the post World War Two American culture from which be-bop and Coltrane emerged, Amiri Baraka's enthusiastic promotion of Coltrane, avant-garde jazz generally, and, finally, a Black Arts era ideology of using art as a weapon, for which Early highlights poems by Baraka, Sonia Sanchez, Haki Mudhubuti, and others as examples.

This final explanation is reinforced by Howard Rambsy, who in the "All Aboard the Malcolm-Coltrane Express" chapter in *The Black Arts Enterprise and the Production of African American Poetry*, writes that poets of the 1960s and 1970s were engaged in a collective project where poets, "memorializing deceased historical figures as black exemplars . . . expressed their political and cultural allegiances", and "In the process of paying homage to jazz musicians . . . accentuated the rebellious, nationalist, and transformative spirits of the music" (Rambsy 2013, p. 102). Rambsy continues, arguing that "the frequency with which these two men [Malcolm X and John Coltrane] were alluded to in black poetry suggest that writers had arrived at a tacit agreement about the value of making these extraordinary cultural figures the subject of their poems" (p. 103). An examination of Coltrane poems by Black Arts era writers displays the veracity of Rambsy's assertion, as the musician was written often as either embodying a Black Arts nationalism with its accompanying revolutionary impulses or as representing a Christ-like or Eastern-redemptive spirituality (or sometimes both within the same poem).

To the first impulse, despite being famously tight-lipped regarding the Civil Rights Movement and subsequent social and political movements (with the exception of composing and recording "Alabama" shortly after the infamous 1963 16th Street Baptist Church bombing), various poets nevertheless transfigured Coltrane into a Black Nationalist political and cultural freedom fighter in their work. Sascha Feinstein points out in "From Alabama to A Love Supreme: The Evolution of the John Coltrane Poem", that throughout his career Coltrane made no direct statements about his association with the Civil Rights Movement (Feinstein 1996, p. 1). Nevertheless, some poets of the 1960s and 1970s adopted Coltrane's sound as the musical embodiment of Black Nationalism in the United States. As Rambsy points out, "Trane's statements were primarily wordless, therefore representing his ideas required poets to delve into the discourses of music in order to reproduce convincing poetic interpretations of what the saxophonist was sharing with his audience" (Rambsy 2013, p. 117). While Amiri Baraka, Sonia Sanchez, Jayne Cortez, Larry Neal, Askia Muhammed Touré, A.B. Spellman, Quincy Troupe, and others used their poetry to promote an image of Coltrane as a cultural prophet, to varying degrees, they also used the innovative sounds of Coltrane's work to complement and deepen their message.

A number of examples of this can be found in Feinstein's "From 'Alabama' to 'A Love Supreme', The Making of a John Coltrane Poem", where he quotes William Harris discussing Coltrane's influence on Amiri Baraka's poetry, "Coltrane takes a weak western form, a popular song ["Naure Boy" from 1965's "The John Coltrane Quartet Plays"], and murders it; that is he mutilates and disembowels this shallow but bouncy tune by using discordant and aggressive sounds to attack and destroy the melody line. The angry black music devours and vomits up the fragments of the white corpse" (Feinstein 1996, p. 2).

Feinstein (and Harris) then identify analogues to Coltrane's music with the tonal shifts that can be found in Larry Neal's "Orisha's", Haki Madhubuti's "Don't Cry Scream", and other works form the era. Kimberly Benston, in *Performing Blackness: Enactments of African-American Modernism*, helps elucidate Feinstein's point, writing, "All the poets . . . felt in Coltrane's music the self-commitment to an exalted state, the 'will' to pass beyond apparent limits of material existence or mere method. Listening to Coltrane—whose revolutionary approach to bebop and the blues changed the nature of the African-American musical lexicon- they sensed a fresh, renovated art no longer derived from the dictates of an inherited tradition but reflected instead the spiritual unity of sacrifice innate to African-American vision. Since this vision was inimical to existing structures, the traditional artistic forms would not be entirely adequate to contain it, and new forms, expressing the radical approach to the old, would necessarily arise" (pp. 120–21).

A few examples from prominent poets from the Black Arts era can help illustrate this re-imagining of Coltrane, as a symbolic figure, while creating a poetic form that throws off the strictures of an "inherited tradition". For example, in Sonia Sanchez's 1970 "a/Coltrane/poem", the poet ties Coltrane's musical innovations, rendered as a repeated "screech" to political and social upheaval, writing, "u blew away our passsst/and showed us our futureeee/screech screeech screeeeech screech/ . . . a lovesupremealovesupremealovesupreme for our blk/people. /BRING IN THE WITE/MOTHA/fuckas/ALL THE MILLIONAIRES/BANKERS/ol MAIN/LINE/ASS/RISTOCRATS (ALL/THEM SO-CALLED BEAUTIFUL/PEOPLE)/WHO HAVE KILLED/WILL CONTINUE TO/KILL US WITH/THEY CAPITALISM" (Sanchez 1991, p. 184). However, the destructive imagery of the poem is tempered at its end, when Coltrane is transformed from a harbinger of destruction to one of redemption and possibility. She writes, "john coltrane./my favorite things is u./showen us life/liven./a love supreme./for each other/if we just lisssssSSSTEN" (Sanchez 1991, p. 186). The "screech" of Sanchez's poem, which, according to Benston, "links thematic transgression with textual transformation" (p. 156), can be compared to Haki Madhubuti's 1969 *Don't Cry, Scream*, where once again, Feinstein can be helpful, writing that the poet "takes his cue from Coltrane's music . . . and interprets the sound as a rebellious holler" (Feinstein 1996, p. 4). While Sanchez and Madhubuti tie Coltrane and his innovative sound to cultural turmoil and revolutionary hopes through "screeches" and "hollers" respectively (Madhubuti 1996, p. 120), other poets emerging from the Black Arts era position the artist as a model figure of healing and hope.

This alternate portrait of Coltrane as muscular prophet and harbinger of positive social change is evoked by Jayne Cortez in 1969's "How Long has Trane Been Gone", where she refers to Coltrane as the "true image of black masculinity", a paternal figure whose "warm arm of music/like words from a Father" will bring comfort to Black people as they "rip those dead white people off your walls" (Cortez 2004, p. 2037). Here, Cortez elevates Coltrane into myth, an impulse shared by Askia Muhammad Touré, who dedicates "Juju" to "John Coltrane, Priest-prophet of the Black Nation" (Touré 1996, p. 171) and writes that Coltrane is, "He Priest Prophet Warrior call and Clarion Call/of essence-US, BLACKNESS" (p. 173). This image of Coltrane is similar to a rendering by Quincy Troupe, who in 1984 writes in "Ode to John Coltrane" that, "With soaring fingers of flame /you descended from Black Olympus /to blow about truth and pain; yeah, /just to tell a story about Black existence" (Troupe 2022, p. 134). Troupe's poem removes the figure of Coltrane from earthly shackles, as he is imagined "Hurtling thru spacelanes of jazz /a Black Phoenix of Third World redemption" (p. 134), who, later in the poem, is also Christ-like, repeatedly referenced as "JC" (p. 135) and also as "John the Baptist" (p. 139), while Coltrane's Eastern spiritual exploration is simultaneously acknowledged by referencing his adopted Sanskit name "Ohnedaruth" (p. 139) (meaning "compassion", and also the title of a song by Alice Coltrane). These are just a few examples of the ways that poets of the Black Arts movement have created the figure of John Coltrane as revolutionary, as myth, as father, as Black Christ, or as transcendent Eastern-gazing mystic, in their own desired image.[1]

Amiri Baraka's multiple portrayals of John Coltrane demonstrate the varying ways that poets "created" Coltrane in their work, as the poet portrayed the jazz great in numerous

iterations over the span of his career. Baraka was an early fan of Coltrane, championing him in reviews in *Downbeat*, writing about Coltrane in *Blues People* as "this new generation's private assassin" (Baraka and Jones 1963, p. 228), and composing the liner notes for 1964's *Coltrane Live at Birdland*. However, it is his poetic re-imagining of Coltrane that concerns us here, and tracing some of Baraka's poetic treatments of Coltrane can shed light on the malleable mythic figure of the musician.

One of Baraka's early Coltrane poems is 1968's "The Evolver" (one of the 99 names of God in the Qur'an, and a reference to Coltrane's musical and spiritual evolution as well as Baraka's own conversion to Islam). This poem, which appeared in the September–October edition of *Negro Digest*, is a celebratory rendition of Coltrane's musical and spiritual development. The first stanza begins, "The power of John Coltrane/The power of God" and ends with "The power of John Coltrane/The worship of God" (Baraka 1968, p. 58), creating an equivalency between the musician and the divine, (an impulse reaching its logical conclusion in the St. John Coltrane African Orthodox Church of San Francisco), before moving to invoke Coltrane's late career musical and Eastern-looking religious explorations, referencing, "the great central stillness to all being . . . "John Coltrane/the emptiness and silence of absolute stillness . . . manifest in the middle of the wrapped around flailing" (p. 58).

What is fascinating about this poem, with its Zen Buddhist focus on silence and stillness, is that it that directly references and, importantly, celebrates, the final 1965–1967 "Om/Interstellar Space" stage of Coltrane's career, as seen in the final line of the poem, which ends "& Pharoah is your Sun" (p. 59), a reference to saxophonist Pharoah Sanders, who, as Albert Ayler reckoned, was the Son of Coltrane the Father in a musical Holy Trinity, of which Ayler is the self-professed Holy Ghost (Kofsky 1968). The John Coltrane of this poem, like the music in the final metamorphosis before his death, is almost transcendent, and Baraka seems to realize this, as the focus on stillness and silence in this poem is echoed by Gerald Early, who asked, referring to Coltrane's late recordings, "where could this music go except into silence?" (Early 1999). According to Baraka, in this poem, silence and stillness are the point and inevitable end, and it is a silence that potentially speaks volumes, as in the sacred sound "Om", "the essence of sound, of breath, of life", or in Coltrane's own terms, "something that hasn't been played before" or the "first vibration—that sound, that spirit which set everything else into being" (Coltrane 1968), which can bring liberation.

Baraka's rendition, here, of Coltrane as Eastern mystic would not last, however, as Baraka's own ideology demanded that Coltrane represent something more fitting to Baraka's evolving revolutionary spirit. In a 1977 interview with Kimberly Benston, Baraka states: "There's a big difference between, say, the Coltrane of 'Giant Steps', and the post-'Love Supreme' Coltrane, when he starts going off into Eastern cosmology and other esoteric ideas, and actually loses a lot of the tough street sound, you know, the fast rhythm, and goes into a contemplative quietist form which loses the fire of actuality—what basically goes on is bourgeois navel watching" (Benston and Baraka 1978, p. 312). The reassessment is important, here, as one potential avenue of Coltrane's persona, that of an Eastern looking spiritual seeker, is effectively blocked by Baraka and subsumed by another version of Coltrane, as evident in his 1979 poem, "Am/Trak".

In this, probably the best known of Baraka's Coltrane poems, the poet presents a biographical and musical progression of Coltrane's life in a five-part sequence, beginning his biography with "a Philly night club/or the basement of a cullet church" (Baraka 2014, p. 197) and moving through "the navy/the lord", his apprenticeships with Miles Davis and Thelonious Monk, and, finally, his recordings of *Meditations*, *Expressions*, and *A Love Supreme* (Baraka 2014, pp. 197–203). However, the musical biography composed by Baraka is incomplete. In the final section of the poem, Baraka centers Coltrane's influences as "slavery, renaissance, bop Charlie parker" (p. 201) and names Coltrane's best-known band, "the inimitable four" (p. 202) of Coltrane, Jimmy Garrison, McCoy Tyner, and Elvin Jones, while omitting the last-stage "bourgeois navel gazing" Eastern-looking Coltrane, and the music he produced. Thus, we are left with an incomplete portrait, one that asserts, "Trane was the spirit of the 60s, He was Malcolm X in New Super Bop Fire" (p. 202) and, despite

his death, continues to be a revolutionary guiding spirit, as the poet states, "And last night I played *Meditations* and it told me what to do/live you crazy mutherfucker and organize yr shit as rightly burning" (p. 203). Here, the poet aligns themself with Coltrane's spiritual directive to "purify" one's self,[2] and takes it a step further, determining to organize in some vague, though one can assume, socially active, rather than strictly spiritual end.

The remaking of Coltrane in Baraka's vision happens throughout the poet's career. Coltrane's name and image are invoked repeatedly, as in 1982's "In the Tradition", where "Coltrane" is shouted as a weapon (Baraka 2000, p. 306), in 1995's "Wise, Why's, Y's", where Coltrane is linked to "The Wise One", (Baraka 2000, p. 485) and "I Love Music", which is dedicated to Coltrane and begins with Baraka quoting Coltrane saying, "I want to be the force that is truly, for good" (Baraka 2014, p. 558), before portraying Coltrane as physically dead, yet symbolically alive, and able to help destroy capitalism, while simultaneously bringing forth limitless positive possibilities, articulated here through the repetition of "can be" (p. 558). The poem alludes to earlier Coltrane works such as *A Love Supreme, Afro Blue, Alabama*, *and My Favorite Things*, before moving to later work such as *Ogunde*, ultimately presenting a holistic vision of Coltrane's musical journey in a single poem: "Everything together/Wailing in unison/A terrible coolness!" (p. 559).

Although Baraka would revisit the figure of John Coltrane repeatedly throughout their career, it is Michael Harper, of all the poets who adopted Coltrane as a muse, who most deeply enmeshed themself and their poetry with the musician, deliberately positioning themself at the intersection of history, poetry, and jazz, with Coltrane as their primary muse.

From the inception of Michael S. Harper's prolific career as poet, essayist, and educator, Harper would repeatedly draw inspiration from the figure of Coltrane and his music. In his 1983 essay, "Don't They Speak Jazz", Harper locates his initial interest in the jazz giant to his childhood exploration of his parents' record collection and a penchant for (permissionless) solo sojourns on New York's subways, coupled with a seemingly preternatural gift for association:

> On that fateful day I was illegally riding after school, and passed my father as he went to work; I knew he'd seen me, though he never let on, and I decided to get on the next train and continue riding. At the next express stop I got off intending to turn around and go back home to the inevitable whipping when I heard a tapping on the window of another train—it was my grandmother; she waved faintly with a hint of a smile. Music and trains! Coltrane. One learns most by getting caught doing the things you love; it leaves an impression. (Harper 1983, p. 3)

In "Harper and Trane: Modal Enactments of *A Love Supreme*", Kimberly Benston writes that the above anecdote marks, for the young poet, the "juncture of music and trains, expression and movement, which he [Harper] names 'Coltrane' . . . [and] arises as the intersection of instruction, proscription, and exploration that transmutes generational difference into familial and cultural transmission. Above all, it is a transformative conjunction, enabling young Michael's passage from eager *tabula rasa* to maturing *ephebe* capable of sustaining the complex tasks of citizenship and artistry by receiving the 'impression' of its contradictions" (Benston 2016, p. 40).

Thus, this early "impression" inaugurated a career that would witness Harper poetically exploring and engaging with a virtual who's who of jazz greats. As they noted in *Arts in Society*:

> I am connected to Coltrane, to Charlie Parker, to Billie Holliday, to Bessie Smith, to Duke Ellington, Louis Armstrong and to all the master musicians who operate in our tradition, to expand it, carry it on, refine it, enliven it, and make it consistent with the aspirations, the human aspirations, of the people in the particular context in which they live, where the music is a vibrant kind of exponential factor directed toward their desires, toward their dreams, their visions of themselves as irreducible spirits. (Chapman 1974, p. 469)

Indeed, a perusal of Harper's oeuvre will demonstrate a continuous engagement with jazz and blues performers such as Charlie Parker, Miles Davis, Elvin Jones, and Billie Holiday, to name only a few. However, despite the poet's embrace of these many artists and their music, it is to Coltrane that Harper would return time and again; from their pioneering National Book Award nominated *Dear John, Dear Coltrane* (1970), and continuing through their final decade, Coltrane remained, for Harper, a fellow traveler with whom they could explore personal, as well as national, racial, and social histories.

In "Trane Time: The Geo-poetic Frequencies of Michael S. Harper", Michael Antonucci notes that, "focusing on the saxophone player's sonic innovations and legendary intensity, Harper's poetry presents Coltrane as an artist, kinsman, freedom fighter, witness, confident and visionary messenger" (p. 57). It is worth noting that Antonucci recognizes, through these descriptors, that Harper wrote a version of Coltrane that was different from the dual Christ-like, militant, messianic figure that he is rendered as in poems by Amiri Baraka, Sonia Sanchez, Jayne Cortez, Haki Madhubuti, Askia Muhammad Touré, and other poets of the Black Arts Movement mentioned earlier. Instead, Harper portrayed Coltrane as a more humanistic symbol, embedded with and embodying a history of African-American art, familial ties, diasporic longing, suffering, and love.

This "humanistic Coltrane" can be seen from the outset of Harper's career, beginning with their 1970 book, *Dear John, Dear Coltrane*. In "Brother John", the poem that opens the volume, Coltrane is celebrated along with Charlie Parker and Miles Davis as embodying a form of black manhood, as the poet repeats, after positing Coltrane within the *Alabama* to *A Love Supreme* part of his career, variations of "I'm black, I am, I'm a black man" (Harper 1985, p. 4). The celebration in this poem is tempered, however, by others in the book, including "Dirge for Trane", which recounts the sense of loss the poet felt when hearing about Coltrane's death, lamenting "the unbrung melody/killed in your brain" (p. 51), and, finally, in the book's eponymous poem, which will be "a restoration against the diasporic distances and alienating wounds imposed against the speaker and Coltrane" (Benston 2016, p. 172).

"Dear John, Dear Coltrane", like Baraka's "Am/Trak", provides a brief biographical sketch of Coltrane, and is framed with the repeated chant of "a love supreme", echoing the singing by Coltrane on that album. Added to this, is a church-like call and response that, when coupled with the repetition of "a love supreme", casts Coltrane as a tragic figure who died for humanity, a seeker "pumping out" tenor kisses and "tenor love" for the rest of us, aching, "for a song you concealed" (p. 75). What distinguishes this poem from some of the earlier work discussed, is, despite the Christ-like imagery, the grounded humanity of it. For in *Dear John, Dear Coltrane,* the poet's framing of Coltrane is of the musician as an image of sickness and loss, using music to redeem both himself and others, but always earthbound, without the soaring hyperbole of the portraits presented by some other poets.

This tone is repeated in "Here Where Coltrane Is", from 1971's *History Is Your Own Heartbeat,* which, again, evokes loss as the primary feeling, acknowledging "Soul and Race/Are private dominions", but emphasizing the power of Coltrane's music (again, *A Love Supreme* and *Alabama*, primarily) to give voice to communal black suffering, creating an "anthem to our memories of you, that is both a lamentation ("For this reason Malcolm is Dead/For this reason Martin is Dead") but also hopeful, as seen in the final line of the poem, "in the eyes of my son are the browns of these men and their music" (Harper 1971, p. 37). Again, Coltrane is seen as a giver of a humanistic hope, who can serve as comfort to the restless memory of martyrdom.

These poems are part of a number of important Coltrane poems Harper would write throughout their career such as "High Modes: Vision as Ritual: Confirmation" (from *History is Your Own Heartbeat* 1971), "My Book on Trane", "Driving the Big Chrysler Across the Country of My Birth", "A Narrative in the Life and Times of John Coltrane, Played by Himself", "Peace on Earth" (from *Healing Song for the Inner Ear* 1985), "A Coltrane Poem: 23 September 1998", "9 23 99: Coltrane Notes on the Millenium", and many others. It's interesting to note that in some of these poems, Harper adopts the voice of Coltrane, writing

from the musician's perspective (as in the aforementioned poems from *Healing Song for the Inner Ear*, further "humanizing" the jazz legend, as seen in "Peace on Earth", in which Harper writes in Coltrane's voice: "I pursued the songless sound/of embouchures on Parisian thoroughfares . . . /how could I do otherwise,/passing so quickly in this galaxy" (p. 192). This image of Coltrane, recognizing the transitory nature of existence, grateful for his spiritual evolution, is, again, an idealized version of the musician, but one that Harper elevates as a symbol of artistic and spiritual redemption following hardship and struggle.

A similar theme can be seen in Harper's 2006 poem, "Faculty Study #421: Brown University Library", where they reflect:

> Without Charles Churchwell[3]
> there would have been 'no study'
> I put Coltrane up on the wall
> the difficulty of the soprano
> a legislation to be passed
> in Providence Plantations
> and could not "sing" on command:
> carrying the mace uphill

In this poem, the hanging of the Coltrane portrait in the study area is a declaration and reclamation by Harper of creative and cultural resilience within an environment built with the profits of slavery. Despite the university's ties to slavery, Harper nevertheless finds, in a library refuge created by a Black man, that, partially due to the image of Coltrane:

> I can forgive anything
> forget nothing
> in the annals of Slavery
> this university is built on (Harper 2006)

Seen here, nearing the final decade of his life, it is Coltrane, more than Charlie Parker, Bud Powell, Billie Holiday, Bessie Smith, Duke Ellington, or any of the other jazz greats about whom Harper writes, whose portrait they hang on the wall, who is the most powerful and meaningful muse to the poet.

It is interesting to note, however, when considering Harper's engagement with John Coltrane, that, early on in Harper's career and artistic formation, it seemed like the poet may have considered Miles Davis an equal to Coltrane, both musically and as source of inspiration. The second poem of *Dear John, Dear Coltrane*, entitled "Alone", is a spare three lines, and posits Davis in transcendent, almost ethereal terms: "A friend told me/He'd risen above jazz./I leave him there", and is prominently dedicated "To Miles Davis" (p. 5). In the aforementioned "Don't They Speak Jazz", from 1983, Harper describes their composition of this poem, writing that while at the Writer's Workshop at the University of Iowa: "My first and only poem on the worksheet in the poetry class was a poem dedicated to Miles Davis, 'Alone', which I've since cut to three lines; 'A friend told me, he'd risen above jazz I leave him there'. It was my bible. How would it be to solo with that great tradition of the big bands honking you on? Could I do it in a poem?" (p. 4). Thus, as a guiding "bible", and emblematic poem, "Alone" is, at the time of "Don't They Speak Jazz", an important declaration by Harper, reflecting on early efforts seeking their individual voice, rooted in an African-American musical and cultural tradition, that, they indicate, would propel their poetry from this point forward. Harper complements this assessment of the poem in the *Notes to the Poems* section of *Songlines in Michaeltree* (2000), where they explain, "'Alone' was conceived as a ballade and shortened to an aphorism. Miles Dewey Davis is a thematic icon in the poet's development" (p. 375).

The passage of time can lead to one reassessing one's work, however, as the edits around "Alone" in *Songlines in Michaeltree: New and Collected Poems* clearly indicate. In this 2000 version of the poem, the importance of Miles Davis as a symbol and a "thematic icon",

hovering "above jazz", is diminished, subsumed by the centrality of Coltrane-as-muse in Harper's work.

In 1970's *Dear John, Dear Coltrane*, "Alone" appears as the second poem in the book, following "Brother John", which is dedicated to John O. Stewart. Following the three-line poem, the dedication "for Miles Davis" (p. 3) is prominently displayed and italicized. A comparison between the poem in 1970 and its placement and a significant edit in the 2002 version, shows a marked elevation of Coltrane-as-muse in Harper's consciousness.

This can be discerned on the title page of *Songlines'* first section, announcing that the following poems are "From *Dear John, Dear Coltrane* 1970". Here, "Alone", formerly dedicated to Miles Davis, is instead presented under the "Dear John, Dear Coltrane" title heading of the collection, diminishing the centrality of Davis. By transposing that accolade ("Alone"), inherent in the poem to Coltrane, Harper further raises the profile of Coltrane in lieu of Miles Davis. Likewise, the dedication "for Miles Davis", has been subsequently removed from the *Songlines in Michaeltree* version of "Alone" (p. 3), further indicating that Coltrane has emerged as the dominant muse in the poet's life and imagination.

Michael Antonucci, writing in "Trane Time: The Geo-poetic Frequencies of Michael S. Harper", notes that "Through *Songlines*, Harper assists critical efforts to trace the trajectory of his poetic project by including an afterward and a set of reader's notes for volume of new and selected poems" (Antonucci 2002, p. 55). While Harper's note regarding "Alone" here doesn't shed significant light on the thinking behind the editorial changes that were made between the 1970 and 2000 versions of the poem, it does help to trace the trajectory of Harper's thinking regarding which jazz great would be their pre-eminent muse.

The editorial changes reflect that in looking over the 30 years of publication history between the two volumes, Harper determined to make it abundantly clear that it was John Coltrane, more than Miles Davis, Charlie Parker, or anyone else, who was their poetic North Star. In the "Notes on Form and Fictions" section of *Songlines*, Harper writes, "A large part of my sensibility has been attracted to the interlopers, outsiders, systemic *makers* who have spent their mature lives producing the *force* that activates the material world with aesthetic energy and a new imagery of possibilities . . . I have tried to pay homage to those beacons of light that made living in the world a little easier" (Harper 2000, p. 373). For Harper, if Miles Davis was once a muse possibly equal to Coltrane, the volume of Coltrane poems, and their assertions in interviews and essays since the publication of *Dear John, Dear Coltrane*, leave no doubt as to which jazz great looms largest in their poetic consciousness.

In a 1990 interview with Charles Rowell, Harper states, "I guess Coltrane was my Orpheus . . . I could itemize my assumptions about Coltrane's effect on me, but he stands as a banner for an attitude, a stance against the world . . . Coltrane is my personal signature for competence and rigor. Also my image for a standard of manliness in the arts, complex, humane, sometimes strident, dense, with great heart and great soul" (Rowell 1990, p. 791). Whether the editing around "Alone" is a result of revisionism, as Harper retroactively realized that it was Coltrane, rather than Davis, who was their "bible", an attempt to make it clear to readers that Coltrane was always their primary muse, or some other rationale,[4] the editorial changes made to, and around, "Alone", give further credence that for Harper and their art, Coltrane-as-muse was a force of energy and possibility; one with which they were able to improvise to bring their vision to the world.[5]

**Funding:** This research received no external funding.

**Institutional Review Board Statement:** Not applicable.

**Informed Consent Statement:** Not applicable.

**Data Availability Statement:** Not applicable.

**Conflicts of Interest:** The authors declare no conflict of interest.

## Notes

[1] This desire to create Coltrane as a poetic metaphor and symbol extends far beyond the Black Arts Movement. As newer generations of poets have continued to mythologize and engage in the legacy of John Coltrane, no clear image or pattern seems to emerge. In *Jazz Poetry from the 1920s to the Present,* Feinstein briefly examines the varied ways that poets beyond the Black Arts have continued to "create" John Coltrane as "the Holy Spirit . . . the music itself" (p. 197), as in Shiraishi's (1991) "Dedicated to the Late John Coltrane", whose love and music "could chane into the wings of a blue angel" (p. 66), as in Joy Harjo's "Healing Animal" (Harjo 1996). Coltrane takes on an ever more contemporary cast in Sean Thomas Dougherty's 2018 "Biography of LeBron as Ohio", which compares a high-school aged LeBron James as both "Baraka on the court" and "Coltrane gold-tones, a kind of running riff" (Dougherty 2018).

[2] In the liner notes of *Meditations,* Coltrane writes, "There is the need to keep purifying these feelings and sounds so that we can see what we've discovered . . . but to do that at each stage, we have to keep cleaning the mirror" (Coltrane 1966), liner notes *Meditations*).

[3] Dr. Charles Churchwell (1926–2018) was a prominent African-American author, educator, and librarian. They were the author of numerous works, including *The Shaping of American Library Education.* Over their career, they held numerous positions at various libraries, including acting as director for the Brown University Library.

[4] In an email to the author regarding the change to "Alone," novelist Rachel Harper, Michael Harper's daughter, wrote: "As for the dedication specifically, my father had a habit (perhaps we could say style) of dedicating poems to people he felt either inspiured the poem, or who he thought needed to hear the message of the poem- so in that way the dedicatoin for "Alone" was a gift for Miles Davis when it was first written, and my hunch is that my father felt it was no longer necessary to present the poem as a gift for Miles. Not that he wanted to take anything away from Miles, but that he was utilizing the power of the poem to introduce the selections from *Dear John*; perhaps he thought the dedication would distract from the focus (of introducing the poems) since the meaning of the poem and the point he's making about leaving someone alone who thinks they are above jazz, is much larger than the dedication to Miles might imply" (Harper 2022).

[5] Harper seems to be leaning towards this ultimate stance, of elevating Coltrane beyond other musicians, as early as 1971's "High Modes: Vision as Ritual: Confirmation", where he writes: "Bird was a mode from the old country;/Bud Powell bowed in modality, blow Bud;/Louis Armstrong touched the old country,/and brought it back, around corners;/Miles is a mode; Coltrane is, power" (Harper 52).

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
