# Peer review of "“Miles Is a Mode; Coltrane Is, Power”: Notes on John Coltrane as Poetic Muse and Michael Harper’s “Alone” in Songlines in Michaeltree (2000)"

_humanities, doi:10.3390/h11050115_

Round 1

Reviewer 1 Report

This short note takes the poem "Alone" and its editorial history as an example to subtantiate the claim that John Coltrane emerged as an important muse for Michael Harper. Miles Davis and other jazz artists also loom large in the poet's works though their influence seems to decrease in comparison to Coltrane's. This point is clearly made and overall convincing.

It is my impression that especially readers who are not intimately familiar with Harper's work might appreciate a bit more context. It would be great if you could expand a little bit more here and there on what exactly this shift tells us about Harper's development as a poet and which functions Coltrane takes on as a muse on different levels - e.g. where exactly can his inspiration be seen and how specifically does his influence change (as already well indicated by the editorial changes to "Alone")? I am well aware that this is a short note and an extensive analysis of Harper's oeuvre lies beyond the scope of the article - however, it could be strengthened if readers would get a more explicit take on the relevance of the observations that are presented and would be able to position them better in the overall assessment of the poet's work and its influences and inspirations taken from jazz music...

Author Response

  1. Additions and changes are in red.
  2. I have expanded the introduction and discussion about how poets, particularly those of the Black Arts era, have “created” John Coltrane in their work. I’ve given a number of examples and taken a close look at the various ways that Amiri Baraka has written about Coltrane.  I’ve looked more closely at work by Sonia Sanchez, Jayne Cortez, Askia Toure, and others to better ground my discussion of Coltrane as a malleable muse for many poets.
  3. I have deepened the discussion of Harper’s engagement with Coltrane, looking closely at moments in his career, as well as additional interviews and essays to clarify and explicate my argument.
  4. I have complemented my discussion with additional theoretical work by Benston, Harper, Early, Feinstein and others working within this space.
  5. Overall, I have better contextualized my article by enlarging my discussion to show that the figure of John Coltrane was an important figure to many Black Arts era poets, and Harper, while also embracing and adopting Coltrane, presented him in a more humanistic way- avoiding much of the strident hyperbole of other poets, while also pointing out that editorial changes in the poem “Alone,” made from 1970’s Dear John, Dear Coltrane to 2000’s Songlines in Michaeltree indicate that over the previous 30 years, Harper elevated Coltrane as his “bible,” above Miles Davis and any other potential muse.

Reviewer 2 Report

This is a timely and insightful article on how Michael Harper’s engagement with the life and work of John Coltrane is central to his poetics. In particular, the author argues that editorial changes and commentary in Songlines in Michaeltree: New and Collected Poems (U of Illinois P, 2002)—situated in the context of Harper’s publication history between the 1970 and 2000 versions of his poem, “Alone”—demonstrate that Harper wanted to make it “abundantly clear” that John Coltrane, not Miles Davis, was his poetic muse (lines 125-126). Although the article is clearly written, and promises to be an important and original contribution to existing scholarship, further development of the argument is needed, especially with regard to presentation of textual evidence from primary sources to support four related claims. First, and most important, there should be a more detailed consideration of Harper's engagement with Coltrane in the poetry composed between 1970 through 2002, although of course it is not necessary to discuss all the relevant poems published during this period. Second, it would be helpful to offer some comparisons and contrasts with at least one or two of the poets of the Black Arts Movement mentioned in the article, given the author's claim that “Harper wrote a version of Coltrane that was different form the dual Christ-like, militant messianic figure he is rendered as in poems by Amiri Baraka, Sonia Sanchez, Jayne Cortez, Haki Madhubuti, Askia Muhammad Toure, and other poets of the Black Arts Movement” (lines 60-63). Third, the author's definition of the "humanistic Coltrane" (line 67) illustrated by poems such as "Brother John," "Dirge for Trane," and "Dear John, Dear Coltrane" needs to be clearer and more specific, possibly with reference to Harper’s “My Poetic Technique and the Humanization of the American Audience,” in Black American Literature and Humanism, edited by R. Baxter Miller (U of Kentucky P, 1981). And fourth, the question raised in the concluding paragraph—namely, whether Harper “retroactively realized that it was Coltrane, rather than Davis, who was his ‘bible'" or whether he attempted "to make it appear as if Coltrane was always his primary muse” (lines 131-134)—should be considered in light of evidence drawn from Harper’s interviews, talks, and critical essays.

With regard to engagement with recent scholarship, the author should cite Kimberly Benston's studies of Harper and Coltrane, such as Performing Blackness: Enactments of African-American Modernism (Taylor & Francis, 2013), and “Harper and Trane: Modal Enactments of A Love Supreme,” Journal of Ethnic American Literature, vol. 6 (2016), pp. 38-60. Adapting some of the theoretical concepts adduced by Benston might also help the author to clarify claims raised in close analyses of the particular poems mentioned above.

Author Response

(The authors gave the same response as above.)

Round 2

Reviewer 2 Report

This manuscript has been sufficiently improved to warrant publication.